# Pathogenesis of Bronchopulmonary Dysplasia: Role of Oxidative Stress from ‘Omics’ Studies

**DOI:** 10.3390/antiox11122380

**Published:** 2022-12-01

**Authors:** Ashley Kimble, Mary E. Robbins, Marta Perez

**Affiliations:** 1Department of Pediatrics, Division of Neonatology, Northwestern University Feinberg School of Medicine, Chicago, IL 60611, USA; 2Ann and Robert H Lurie Children’s Hospital of Chicago, Chicago, IL 60611, USA; 3Stanley Manne Children’s Research Institute of Chicago, Chicago, IL 60611, USA

**Keywords:** bronchopulmonary dysplasia (BPD), oxidative stress, antioxidants, epigenetics, transcriptomics, proteomics, metabolomics, newborn

## Abstract

Bronchopulmonary dysplasia (BPD) remains the most common respiratory complication of prematurity as younger and smaller infants are surviving beyond the immediate neonatal period. The recognition that oxidative stress (OS) plays a key role in BPD pathogenesis has been widely accepted since at least the 1980s. In this article, we examine the interplay between OS and genetic regulation and review ‘omics’ data related to OS in BPD. Data from animal models (largely models of hyperoxic lung injury) and from human studies are presented. Epigenetic and transcriptomic analyses have demonstrated several genes related to OS to be differentially expressed in murine models that mimic BPD as well as in premature infants at risk of BPD development and infants with established lung disease. Alterations in the genetic regulation of antioxidant enzymes is a common theme in these studies. Data from metabolomics and proteomics have also demonstrated the potential involvement of OS-related pathways in BPD. A limitation of many studies includes the difficulty of obtaining timely and appropriate samples from human patients. Additional ‘omics’ studies could further our understanding of the role of OS in BPD pathogenesis, which may prove beneficial for prevention and timely diagnosis, and aid in the development of targeted therapies.

## 1. Introduction

Bronchopulmonary dysplasia (BPD) was first described by Northway et al. in 1967 as a new disease in moderately premature infants resulting from the intersection of prematurity, surfactant insufficiency, ventilator-associated lung injury and oxygen toxicity [1,2]. In the ensuing 50 years, our understanding of the pathogenesis of BPD expanded from direct injury to the immature lung parenchyma to include prenatal and postnatal environmental factors. In 1999, the concept of “new” BPD, a disease state that primarily affects neonates with birth weights of less than 1000 g, was introduced. “New” BPD results from an arrest in lung development that produces histopathological changes, including profound alveolar simplification, dysmorphic microvasculature, and interstitial changes. This is in contrast to alveolar overinflation and fibrosis typical of “old” BPD seen in older, more mature neonates who experienced lung injury secondary to oxygen toxicity and mechanical ventilation [3,4,5,6]. 

BPD is one of the most common complications of premature birth, primarily affecting neonates born at less than 28 weeks of gestation [7]. Oxidative stress (OS) has long been recognized as an important antecedent to the development of many neonatal diseases including BPD. Saugstad introduced the concept linking oxygen radical disease of the newborn and BPD via increased levels of hypoxanthine, a potential free radical generator associated with hypoxic cellular injury, in a 1988 review paper [8]. OS has since been linked to several pathological conditions in neonates in addition to BPD including necrotizing enterocolitis, retinopathy of prematurity, intraventricular hemorrhage, and patent ductus arteriosus [9,10,11,12]. 

Preterm infants who go on to develop BPD have different oxidation profiles compared to infants who are not affected by BPD, with increased markers of OS even in the early stages of the disease [13,14]. In one recent study of serum and tracheal aspirates from very low birth weight infants, infants later diagnosed with BPD had significantly higher 8-hydroxy-20-deoxyguanosine (8-OHdG), a biomarker of oxidative damage, on day 1 and day 28 of life [15,16]. 8-OHdG was also increased in urine samples on day 7 of life, again supporting OS as an important factor in the pathogenesis of BPD [17]. Other markers of OS, such as 3-nitrotyrosine, have been shown to be elevated in plasma samples from preterm infants who go on to develop BPD [18]. Interestingly, markers of OS persist into adolescence in infants born prematurely, as evidenced by higher 8-isoprostane levels in exhaled breath condensate, suggesting prolonged airway dysfunction after preterm birth [19]. 

The reduction of oxygen to water is the foundation of aerobic respiration; however, intermediaries of this reaction can be toxic. Each step generates highly reactive oxygen species (ROS), including superoxide radical (O_2_^−^), hydroxyl radical (OH^●^) and hydrogen peroxide (H_2_O_2_). O_2_^−^ and OH^●^ are highly toxic free radicals that cause molecular damage via lipid peroxidation, protein oxidation, and deoxyribonucleic acid (DNA) damage [20,21,22]. In contrast, H_2_O_2_ is a ROS but not a free radical, and functions as an essential signaling molecule [23]. Reactive nitrogen species (RNS), a subset of ROS, include various compounds derived from nitric oxide such as nitrogen dioxide (NO_2_), peroxynitrite (ONOO^−^), and nitrosoperoxycarbonate (ONOOCO_2_^−^). While RNS at physiological concentrations are essential for normal biological functions, their overproduction can result in extensive tissue injury through the modification of target molecules via oxidation, nitrosylation, and nitration. Common sources of oxidative and nitrosative stress in critically ill newborns include exposure to hyperoxia, mechanical ventilation, and inflammation.

All newborn infants are exposed to relatively high oxygen concentrations as they transition from a low oxygen environment in utero to the high oxygen environment of room air during the first few weeks of life [24]. Newborns with respiratory insufficiency may require supplemental oxygen resulting in additional exposure to hyperoxia. At the same time, many of these infants will likely experience episodic hypoxia due to their underlying lung disease and clinical instability [25]. Critically ill infants may also experience bouts of infections and inflammation, factors which increase ROS via the respiratory burst, a rapid release of ROS during the phagocytosis of microbes. Preterm and asphyxiated infants also have free iron present, which generates ROS via the Fenton reaction (H_2_O_2_ + Fe^2+^→OH^●^ + OH^−^ + Fe^3+^) [26]. All of these factors may result in increased exposure to oxygen toxicity and associated OS from multiple sources in preterm infants.

An endogenous antioxidant system is comprised of enzymes and non-enzymatic molecules to protect against ROS. Antioxidant enzymes include superoxide dismutase (SOD), catalase (CAT), glutathione (GSH) peroxidase (GPX), and glutathione reductase (GSR). To review, SOD, the most powerful antioxidant enzyme, catalyzes the dismutation of O_2_^−^ to H_2_O_2_. CAT then reduces H_2_O_2_ to water and O_2_. GPX is a selenoperoxidase that catalyzes the reduction of H_2_O_2_ to water by oxidizing GSH to glutathione disulfide (GSSG). GSR then reduces GSSG to GSH using NADPH as an electron donor, which is important to maintain the function of GPX [27]. Non-enzymatic antioxidant molecules include vitamins (for example vitamins A, C and E), β-carotene, allopurinol, melatonin, and plasma metal binding proteins such as ceruloplasmin and transferrin [13,28]. These molecules primarily function as free radical scavengers and neutralizers [27]. Homeostasis between ROS production and the antioxidant system is critical; excess production of ROS beyond the neutralizing capability of the antioxidant system results in OS [29]. 

Premature infants have decreased antioxidant mechanisms compared to neonates born at term. Normally, human lung CAT activity increases throughout gestation [30]. Similarly, lung SOD, CAT and GPX increase by up to 200% in the final 10–15% of pregnancy in animal models [31,32,33]. In preterm infants, erythrocyte CAT and GSH activity are reduced compared to term infants [34,35]. Preterm neonates are also unable to increase lung mitochondrial SOD expression in response to hyperoxia exposure [36,37]. Finally, antioxidant molecules (including vitamin A, vitamin E, transferrin and ceruloplasmin) are lower in premature neonates compared to term infants, particularly in patients who later develop BPD or other oxidative diseases of the newborn [14,38,39,40]. Various factors impacting pregnancy and labor may also affect the neonate’s ability to handle OS. For example, maternal obesity may influence the neonatal antioxidant system, including CAT and SOD activity in term infants at birth [41]. Delivery by cesarean section in the absence of labor, has also been linked to reductions in antioxidant potential in both preterm and term infants [42]. For these reasons, premature neonates have an inherently decreased antioxidant capacity to manage an increased exposure to OS. 

The types of OS within the premature lung that may contribute to the development of BPD are multifactorial, including changes in oxygen tension, inadequate defense against OS, insufficient antioxidant response, and inflammation [6,29,43]. These mechanisms, which cause a change in the milieu of the developing lung and alter lung maturation, have been the focus of other review articles [12,44,45]. Our goal in this discussion is to examine the subsequent effects of the altered redox state within this disease on the ‘omics’ of BPD. We will review data from studies of genomics, epigenetics, transcriptomics, proteomics, and metabolomics with a focus on the role of OS in the pathogenesis of BPD. When available, data from cord blood and early-life omics studies with a focus on the possible use of OS pathways as biomarkers and early therapeutic targets are reviewed. By examining these interlinked ‘omics’ (Figure 1), we will examine the cascade from altered gene accessibility to the ultimate metabolism of gene products in relation to oxidative signaling. 

## 2. Genomics

Gene variants associated with BPD occur a priori of OS exposures associated with the development of BPD; however, variants in antioxidant response genes or ROS may confer increased susceptibility for BPD. As genomic material is the antecedent of the downstream ‘omics’ discussed in this review article, genetic variants associated with BPD will be briefly reviewed prior to discussing other related elements such as epigenetic or transcriptomic changes. 

Early twin studies demonstrated a genetic predisposition to BPD [46,47]. The first identified single nucleotide polymorphisms (SNPs) associated with BPD were not within OS pathways but rather in the surfactant protein genes [48,49]. However, using a targeted analysis of genes highly involved in OS (SOD1, SOD2, SOD3, CAT), an Italian group described an association between SNP variants within the SOD2 and SOD3 genes and the development of BPD [50]. SNPs within the NADPH oxidase (NOX) gene have been shown to increase ROS production and are associated with the development of BPD [51]. Interestingly, variants within the nuclear factor (erythroid-derived 2)-like 2 (NRF2) gene, which coordinates over 50 antioxidant response elements, have been associated with decreased incidence of severe BPD [52]. 

Although targeted gene investigations identified OS genes variants associated with BPD, this approach can introduce bias or miss previously unidentified pathways, as opposed to using an “unsupervised” approach that does not depend on prior assumptions or knowledge. The first BPD genome-wide association study (GWAS) identified 70 variants within the SPOCK2 gene (SPARC (osteonectin), CWCY and Kazal-like domains proteoglycan 2) associated with moderate/severe BPD across a racially diverse cohort of 418 infants [53]. Using whole-exome sequencing (WES), Carrera et al. identified a high mutation burden in genes previously associated with BPD, including genes within the Toll-like receptor (TLR)-family (TLR10, TLR1, TLR4), OS-related genes (EPHX2, MTHFR, EPHX1) and surfactant metabolism genes (SFTPD, ABCA3). This analysis also identified other genes associated with BPD, including matrix metalloproteinase (MMP1), nitric oxide synthase (NOS2), C-reactive protein (CRP) and lipopolysaccharide binding protein (LBP) [54]. 

Subsequent GWAS and WES studies have not consistently identified SNPs associated with BPD or been able to validate previously described SNPs. In a GWAS of 1726 very low birth weight and extremely preterm infants, no SNPs were significantly associated with BPD, including previously identified SNPs from 14 previous GWAS [55]. Similarly, the WES for a subset of 146 infants within the Prematurity and Respiratory Outcomes Program cohort, did not identify variants related to OS or ROS genes [56]. In a different group of 834 extremely premature infants, no SNPs reached significance at the genome-wide level; however, secondary gene ontogeny pathway analysis, though unrelated to OS response, did reach significance for severe BPD or death [57].

Analysis of gene sequence variants of antioxidant response elements has not consistently yielded an association with BPD. The NFE2L2 SNP (NFE2L2; rs6721961) was associated with decreased incidence of severe BPD [58]. Even though this study did not reveal any differences in Nrf-2’s target antioxidant response elements, specifically for SNPs of glutamate-cysteine ligase catalytic subunit (GCLC), glutathione S-transferase Pi 1 (GSTP1), heme oxygenase 1 (HOX1), and NADPH: quinone oxidoreductase 1 (NQO1), later studies have revealed SNPs that may have an association with BPD [58,59]. Higher frequency of the inactivating NQO1 SNP (C^609^T), which functions as a highly inducible phase II antioxidant enzyme that catalyzes quinones to hydroxyquinones, was observed in infants diagnosed with BPD, and all infants with a birth weight of less than 1000 g and at least one copy of the SNP later developed BPD [60]. In a study of 659 very low birth weight premature infants, the presence of the homozygous NQO1 SNP (NQO1; rs1800566) was associated with increased BPD [58]. However, an inactivating SNP of GSTP1 (A^313^G) was more prevalent in infants born with a birth weight of less than 1000 g who were later diagnosed with BPD, though the sample size was quite limited [59]. The substitution of CYP2B6 G^516^T (CYP2B6; rs3745274), a member of the cytochrome P450 family with antioxidant roles, was more prevalent in infants diagnosed with BPD and may confer susceptibility [59]. These studies suggest that variations in antioxidant defenses may influence an infant’s susceptibility to develop BPD. 

The genetic influence of race on BPD is also of genomic relevance. The top variant identified by a comparison of the GWAS of several racial groups identified variants within an intron of neuroblastoma 1, DAN family BMP antagonist (NBL1; rs372271081), a gene associated with nitric oxide metabolism. Although this did not reach significance at the genomic level, this variant was protective against BPD. While this variant was most highly correlated with African ancestry, it was protective against BPD within the other racial and ethnic groups included in the cohort as well [61]. In a genetically homogenous Finnish population, the most promising and reproducible SNP (though not significant) was variant rs11265269, located near the CRP gene. This was also present when cross-referenced with French-African and Caucasian ancestry reference datasets [62]. More studies are needed to expand our knowledge of the interaction between racial influences on genomic changes and BPD. 

## 3. Epigenetics

Epigenetics are heritable changes to gene expression that are independent of alterations to the DNA sequence. The mechanisms that cause these extra-genomic changes include DNA methylation, histone acetylation and non-coding RNAs. DNA methylation occurs when DNA methyltransferases (DNMTs) catalyze the conversion of cytosine to 5-methylcytosine islands in CpG islands within the genome. De novo production of methylation is performed by Dnmt3a or Dnmt3b while Dnmt1 maintains methylation between generations of cells [63]. DNA methylation blocks promoter accessibility to the cellular machinery responsible for gene transcription. Histone acetylation alters the nucleosome coiling and changes the chromatin density affecting the accessibility of the genome for gene transcription [64]. Non-coding RNAs are RNA transcripts that are not converted into proteins; these fall into different categories based upon their relative size where microRNAs (miRs) are <22 nucleotides long and long non-coding RNAs (lncRNAs) are >200 nucleotides long. Their primary function is to maintain the post-transcriptional regulation of protein production by disrupting the transcription and/or translation of their target mRNA [65]. Mechanisms linking OS to epigenetic changes have been the subject of other reviews [24,66]. 

DNA methylation naturally changes over the course of murine lung development with 209 differentially methylated and expressed genes, including some genes related to OS such as Mmp3, Sod3 and peroxiredoxin, between the neonatal and adult periods. Dnmt expression naturally increases in murine lungs over the course of postnatal development [67]; however, this increase can be exaggerated in the neonatal period due to hyperoxia exposure [68]. Following neonatal hyperoxia exposure, mice have a threefold increase in total genome promoter methylation, of which one notable pathway, the Creb1 pathway, is downstream from Tgf-β and plays an important role in lung epithelial apoptosis in response to OS, which may lead to dysregulated cell death following OS [69]. Similarly, after exposure to intermittent hypoxia, neonatal rats have an increase in Sod2 promoter methylation which results in decreased mRNA, protein production and enzyme activity [25]. Moreover, epigenetic mechanisms can interact with each other to affect transcription as evidenced by increased methylation of the promoter and correlative decreased expression of the miR-17~92 cluster following a murine hyperoxia model and whose target Tgf-β’s relationship with OS is mentioned above [68]. Moreover, methylation of histone 3 and 4 has been described following a perinatal inflammatory exposure model as well as a rescue of the alveolar simplification phenotype following miR-29b supplementation [70].

Using a commercial microRNA-seq assay to assess the expression of 521 mature murine miRs, one group identified 117 miRs whose expression changed over the first month of postnatal lung development; of these, 66 were hyperoxia-responsive following oxygen exposure with increased expression in 65 of 66 miRs (miR-680 was the only decreased miR). Further computational analysis using TargetScan, a software that associates miR sequences to target mRNA, yielded 152 mRNAs with changes due to miR expression changes [71]. Using an unsupervised approach to examine changes in lncRNA expression following a murine hyperoxia-exposed model for BPD, one group found more than 1700 lncRNAs were dysregulated with a relatively even split between increased and decreased expression [72]. When examining lncRNA, miR and mRNA interactions in a murine BPD model, one group identified differential suppression of 155 miRs, with changes in 455 target genes and 10 overlapping lncRNAs. Subsequent pathway analysis demonstrated significant increases in Hif-1 attributed to altered expression of the various RNAs [73]. 

Due to the known differences in clinical BPD outcomes based on biological sex, animal models have been used to investigate the sex-specific epigenetic landscape following OS. Chromatin immunoprecipitation of the H3K27ac histone modification mark was used to assess sex-related differences in the epigenetic response to hyperoxia in mice. The genetic sexes had distinct patterns regarding histone acetylation, especially regarding genes associated with lung development [74]. Following oxygen exposure, female mice upregulate miR-30a, a miR with targets related to angiogenesis. The female mice have greater changes in gene expression for those angiogenic target genes, especially the Delta-like 4 (Dll4) gene, which is involved in the Notch signaling pathway, compared to male mice. This effect is sustained and increased after recovery from oxygen exposure; this is proposed as a possible mechanism to explain why female mice have better-preserved pulmonary vascular development compared to males following hyperoxia [75]. 

Since preterm infants often receive supplemental oxygen during resuscitation, Lorente-Pozo et al. examined the effects of oxygen load during resuscitation on methylation for infants born at less than 32 weeks gestation. In this study, an oxygen load greater than 500 mL O_2_/kg of birth weight was associated with significant loss of methylation at 2196 CpG sites and an increase in methylation at 430 CpG islands. Genes with decreased methylation were associated with pathways for genomic integrity, DNA damage checkpoints, cell adhesion, cell cycle progression and mitochondrial electron transport that are likely involved in processing free radicals and ROS [24]. Global methylation studies on buccal swabs during the week of discharge did not demonstrate any methylation changes in ROS genes or antioxidant enzymes [76]. Epigenome-wide association studies on cord blood samples found differential hypomethylation and hypermethylation of CpG islands in BPD patients, including the SPOCK2 gene and genes within a gene ontogeny pathway entitled production of reactive oxygen species [77]. In a small dataset of 54 preterm infants, Cohen et al. found changes in the histone acetyltransferase binding activity and chromatin remodeling pathways (aka chromatin packaging). Individual genes did not reach significance in these cord blood samples but the pathways did [78].

Examinations of human infant lung tissue typically require using autopsy lung samples, most frequently from the Biorepository for Investigation of Neonatal Diseases of the Lung (BRINDL), as accessing tissues in living and surviving infants is challenging. In an unsupervised global methylation analysis of such autopsy samples, there were 149 differentially methylated genes including the antioxidant enzymes, glutathione S-transferase M3 (GSTM3) and SOD3, in infants with BPD compared to infants without the disease [67]. Similar to the murine studies, targeted miR investigations identified several significant epigenetic changes along the expression pathway of the miR-17~92 cluster from infants who died of severe BPD including increased promoter methylation, increased DNMT expression and 80–90% decreased expression of mature miRs within the group [79]. Similar confirmatory studies regarding miR-30a (discussed above in the murine studies) found decreased miR and increased DLL4 target in patients who died of severe BPD [75]. One way to overcome the difficulties in obtaining neonatal lung tissue is to indirectly infer the lung environment from sputum samples. Using this approach, Lal et al. examined the miRs present in bronchoalveolar lavage fluid (BALF) exosomes and found that the overall quantity of exosomes was decreased in BPD patients with 40 differentially expressed miRs in these samples [80]. These studies provide a unique ability to look at the epigenetic milieu in disease survivors.

Within a large GWAS study from blood samples of infants with severe BPD, the miR-219 pathway reached significance with correlative expression changes in 32 target genes (14 increased and 19 decreased). Further investigations suggest these disparate effects on target genes may be related to alternate splicing [57]. Follow-up murine confirmatory studies found miR-219 to regulate platelet-derived growth factor receptor alpha, which is essential for normal lung development and independent of OS pathways [81]. An examination of the cord blood exosomes for differential miR and lncRNA expression found that 135 and 104 changes in expression, respectively, are associated with the subsequent development of BPD, although, specific OS pathways were not identified in this patient cohort [82]. Another group found 328 upregulated and 90 downregulated miRs within cord blood exosomes to be associated with the development of BPD; several of these changes overlapped to affect the phosphoinositide-3-kinase–protein kinase B/Akt (PI3K-AKT) signaling pathway, which is involved in cellular metabolism [83]. Targeted investigations regarding the miR-17~92 cluster demonstrated decreased circulating amounts of the mature miRs within the first week of life for patients later diagnosed with severe BPD [68]. These early alterations in epigenetic pathways in cord blood, early BALF or blood samples have the potential to become an early predictor of BPD prior to the 36 weeks corrected gestational age currently required to diagnose BPD clinically [84].

## 4. Transcriptomics

The transcriptome represents the first stage of gene expression and is the complete set of RNA transcripts produced by the genome. The transcriptome is commonly evaluated by studies including genome-wide gene expression assays and RNA sequencing. Transcriptome studies of neonatal animal models exposed to hyperoxia have identified differentially expressed genes involved in ROS metabolism or redox balance, in addition to other pathways including DNA repair, cellular proliferation, angiogenesis, inflammation, and lung development [85,86,87]. One recent transcriptome study of single cells in neonatal mice demonstrated increased expression of the cell cycle inhibitor Cdkn1a in response to hyperoxia, which may potentially protect the lung against OS [88].

Various transcription factors are impacted by hyperoxia exposure. Activation protein 1 (Ap-1) is involved in oxygen detoxification, and its Fos and Jun members are activated in a premature rat model in response to hyperoxia [89]. Cytochrome P450 1A1 (Cyp1A1) may attenuate hyperoxic lung injury via the aryl hydrocarbon receptor (Ahr) pathway, a regulator of hyperoxia-associated gene expression in the neonatal murine lung [90,91]. One study demonstrated Ahr pathway upregulation during acute hyperoxia exposure at 24 h, though this did not persist at 48 h [92]. Nrf2, an essential transcription factor involved in antioxidant response and defense that regulates over 50 genes with antioxidant response elements, is upregulated in response to hyperoxia in multiple animal models [86,93,94]. Identified targets of Nrf2 include solute carrier family 7 member 11 (Slc7a11), glutathione peroxidase 2 (GPX2), and macrophage receptor with collagenous structure (Marco) [91]. Slc7a11 encodes xCT, a key component of the high-affinity cysteine/glutamate exchange transporter system that mediates cellular cysteine uptake for GSH synthesis and the maintenance of GSH levels [95]. Interestingly, Slc7a11 is upregulated in the acute initiation phase of hyperoxic lung injury of neonatal but not adult mice [94,95,96]. Additionally, Nrf2 deficiency causes hyperoxia-induced lung inflammation during the saccular phase of lung development in mice [95]. Together, these findings suggest a critical role of Nrf2 in the management of OS. 

Several antioxidant enzymes are upregulated in response to hyperoxia exposure. During the saccular and alveolar developmental phases of neonatal mice, oxidoreductases are increased, including GPX2, GPX3, GSTM1, GSTM2, GSTM3, glutathione S-transferase alpha 3 (GSTA3), GSTA4, and microsomal glutathione S-transferase 1 (MGST1) [87]. A highly inducible phase II antioxidant enzyme that catalyzes quinones to hydroxyquinones, NQO1, is upregulated during the acute initiation phase of hyperoxia in mice [95]. Other enzymes including SOD2, thioredoxin reductase 1 (TXNRD1), and sulfiredoxin 1 (SRXN1) are also upregulated in hyperoxia-exposed animals [86]. A few antioxidant molecules have been identified to be impacted by hyperoxia exposure. Metallothionein-1 (Met-1), which scavenges OH^●^ and O_2_^−^ radicals, is increased in hyperoxia-exposed rats between birth and day 10 of life, in contrast to the decrease in Met-1 levels in controls during this same time interval [89,97]. Other studies have examined the effects of knockouts of antioxidant enzymes on the transcriptome. For example, GSR deficiency results in a different hyperoxia response in neonatal mice with enhanced upregulation in the antioxidant thioredoxin system (NADPH, TXNRD, thioredoxin) in the lung [98]. 

Since male gender is considered an independent risk factor in the development of BPD, a few studies have investigated sexual dimorphisms in pulmonary transcriptome changes in response to hyperoxia. Despite sex-specific regulation of 585 genes in males and 327 genes in females during the acute cellular activation phase of hyperoxic lung injury in neonatal mice, genes involved in ROS metabolism, including Met, Hox1, Gsr, Slc7a11 and Nqo1, were similarly regulated in both male and female mice [95]. Interestingly, Nrf2, which has antioxidant response elements, is upregulated in female mice but downregulated in male mice during the acute and chronic phases of hyperoxia exposure (on day 7 and 21 of life, respectively), suggesting sex-specific modulation of this pathway [85]. These studies suggest sexual dimorphisms involving ROS metabolism may exist within the pulmonary transcriptome. 

Only a few studies have evaluated the human transcriptome in patients with BPD. These studies are generally limited by a small number of patients with difficulty in obtaining appropriate samples, which limits generalizability. Sahni et al. did not identify any differentially expressed genes involved in redox balance in 90 patients at 36 weeks postmenstrual age with BPD compared to control patients by multi-spot enzyme-linked immunosorbent assays [99]. A study of peripheral blood samples from 21 infants born at less than 29 weeks gestation age taken in the first week of life and again and at the time of BPD diagnosis revealed 431 differentially expressed genes, with upregulation of genes involved in ROS detoxification in addition to red blood cell development and oxygen transport, and downregulation of immune-mediated pathways in patients with BPD compared to control patients [100]. Pathways involved in inflammation, but not ROS metabolism, were increased in lung tissue samples from the autopsy of 11 patients who died with BPD compared to 17 patients who died without BPD [101]. A recent transcriptomic study that compared tracheal aspirate samples from extremely preterm infants with BPD to term infants identified 33 differentially expressed genes, primarily involved in inflammation, cellular growth, and lung development but not redox balance [102]. Further investigations are warranted to determine changes in the human transcriptome following preterm birth, including transcriptomic signatures of cord blood and the pulmonary transcriptome, as these studies may aid in identifying early biomarkers of BPD.

## 5. Proteomics

Additional approaches to identify biomarkers and potential therapeutic targets in BPD have included the use of proteomics, the study of structure and function of protein networks. Proteomic analysis of lung tissue in neonatal mouse models of hyperoxic lung injury has demonstrated upregulation in proteins involved in OS response, in addition to cytokine signaling, neuron apoptotic processes, negative regulation of mitochondrial organization, and epithelial cell proliferation and migration, indicating activation of these pathways in hyperoxia [87]. These proteomic changes correlated with transcriptomic data that demonstrated dysregulation in genes that regulate OS in hyperoxia-exposed animals. 

Proteomics studies may help investigate protein networks involved in disease pathophysiology and progression; however, human studies can be limited by the generalizability of sample types (for example, urine and serum findings may be difficult to apply to the lung). Tracheal aspirates may be the best option for researchers interested in human lung proteomics, though they can only be obtained invasively, making it challenging to analyze profiles from control patients. Only a few studies have used postmortem lung samples. Proteomic analysis of tracheal aspirates from infants later diagnosed with BPD and from control infants, striated by gestational age (23–25 weeks gestational age and 26–29 weeks gestational age), identified distinct protein profiles. Surfactant protein-A2, annexin-3, calcium and integrin binding protein-1, leukocyte elastase inhibitor, chloride intracellular channel protein 1 and calcyphosine were all differentially expressed in infants with severe BPD born at 23–25 weeks gestation as compared to babies born at the same gestation age who did not develop BPD [103]. Additionally, the study found distinct differences in protein profiles between younger vs. older gestational ages and between mild vs. severe BPD. Notably, leukocyte elastase inhibitor levels were lower in patients born between 23–25 weeks gestation compared to 26–29 weeks gestation. Leukocyte elastases are potent proteolytic enzymes that have been previously implicated in the pathogenesis of acute lung injury and exogenous inhibition of these enzymes has been associated with decreased OS in asthmatic rats [104,105,106]. A recent study that looked at the inactivation of alpha-1 antitrypsin (AAT) in tracheal aspirates found evidence of oxidative AAT inactivation in neonates who develop severe BPD, suggesting a role for OS in the development of this disease [107].

The proteome of plasma samples also differs between infants diagnosed with BPD and those without BPD. Cord blood analysis showed significantly decreased afamin, gelsolin and carboxypeptidase N subunit 2 in infants later diagnosed with BPD [108]. Notably, afamin is a vitamin E binding serum glycoprotein involved in OS-related cellular processes such as anti-apoptosis. Although vitamin E is a potent antioxidant and vitamin E deficiency has been associated with an increased risk of BPD, supplementation has not reduced the risk of developing the disease [109,110]. Other proteome studies of plasma samples from infants diagnosed with BPD have shown distinct protein profiles in BPD patients, with decreased galectin-3 binding protein, decreased hemoglobin subunit gamma-1, and increased serotransferrin levels compared to control infants [108]. 

A recent study that analyzed urine samples collected within 72 h of birth validated 20 proteins previously found in blood and tracheal aspirates from BPD infants, including proteins involved in inflammation and abnormal pulmonary development [111]. Overall, urine proteome studies may represent a less-invasive method to study molecular changes associated with development of BPD.

## 6. Metabolomics

Metabolomics is the study of metabolites in bodily fluids and tissues that reflect the tissue’s current physiologic state. It is a dynamic process that can be impacted by many factors including genetic predisposition and environmental factors such as prematurity, oxygen exposure, ventilation, nutrition, and drugs. As such, it represents a snapshot obtained at a specific timepoint, potentially limiting its generalizability. Metabolomic studies have revealed prenatal and postnatal differences in infants with BPD compared to infants without significant lung disease. 

Prenatal metabolic profiling of amniotic fluid from women in preterm labor has demonstrated higher levels of leucinic acid, hydroxy fatty acids, oxy fatty acids, and a sulphated steroid in mothers of infants that were later diagnosed with BPD. Amniotic fluid from those women also had lower levels of S-Adenosylmethionine, a precursor to the antioxidant GSH, and an important intracellular defense against ROS [112]. To date, there have been very few studies investigating metabolomics in the cord blood of BPD patients. One multiplatform metabolomics evaluation of umbilical cord blood plasma noted dyslipidemia on the metabolic profiles of preterm infants who went on to develop BPD and BPD-associated pulmonary hypertension. Several of those lipid metabolites (namely, peroxisome proliferator-activated receptor-gamma and 9-hydroxyeicosatetraenoic acid) have been previously associated with OS [113]. Taken together, these findings suggest that infants with in-utero exposure to OS may be at a higher risk of subsequent BPD development. 

Another approach to studying metabolomics in the BPD population has been to investigate early-life tracheal aspirates. Metabolic profiles of BALF samples at birth, after surfactant administration, and at the time of extubation showed overexpression of ten metabolites (undecane, decanoic acid, dodecanoic acid, hexadecanoic acid, octadecanoic acid, hexadecanoic acid methyl ester, 9-octadecanoic acid, tetracosanoic acid, myristic acid, phosphate) in infants receiving mechanical ventilation after surfactant administration [114]. This may be reflective of an inflammatory response as these metabolite peaks are similar to BALF metabolomic studies of patients with cystic fibrosis [115], though the precise diagnostic and therapeutic importance of these compounds in BPD remains to be determined. A study by Lal et al. investigated the metabolic profile of tracheal aspirates collected within six hours of birth from infants later diagnosed with BPD and compared them to infants who did not develop significant lung disease. The authors identified alterations in pathways that are involved in fatty acid metabolism as well as steroid hormone biosynthesis, which were hypothesized to modify lung development and injury susceptibility. Notably, no pathways specifically involved in redox balance were identified [116]. Another study of tracheal aspirates collected from preterm infants during the first week of life demonstrated 18 significantly different metabolites between BPD and control patients, independent of gestational age and birth weight. Glycine was one of the most significantly increased metabolites in infants diagnosed with BPD compared to control infants. The other highly discrepant metabolites include histidine, glutamic acid, citrulline, isoleucine and two acylcarnitines [117]. 

Urinary samples have also been utilized to identify early-life metabolomic changes in BPD. A study of urine metabolites collected at 24–36 h of life identified five distinct urinary metabolites in premature infants later diagnosed with BPD. Compared with control infants, infants with BPD had increased lactate, taurine, trimethylamine-N-oxide (TMAO) and myoinositol, and decreased gluconate levels [118]. Another study of urinary metabolomics demonstrated increased alanine and betaine and decreased TMAO, lactate, and glycine levels in urine samples collected on day 7 of life from infants later diagnosed with BPD when compared to infants without BPD [119]. Several of these metabolites may be involved in redox balance. For example, glycine is involved in the synthesis of GSH, an important antioxidant. Deficiencies in GSH production have been linked to increased markers of OS in aging, while glycine supplementation has been associated with reduced markers of OS in human subjects as well as in a rat model of sucrose-induced insulin resistance [120,121]. Lactate, another metabolite identified by urine metabolomics, has been associated with decreased formation of ROS by scavenging free radicals [122]. Finally, betaine has antioxidant properties, including protection against lipid peroxidation and maintenance of antioxidant lymphoid tissues [123,124]. 

Additional approaches to study of metabolomics in BPD have included smell prints from tracheal aspirates and fecal samples. Smell prints of organic volatile metabolites of tracheal aspirates on day 7 of life are distinguishable between infants who do and do not develop BPD [125]. Fecal organic volatile metabolites are also distinguishable between infants later diagnosed with BPD and controls on day 14, 21, and 28 of life [126]. While individual metabolites are not identifiable by this mechanism, these findings likely reflect alterations in systemic inflammation, metabolic pathways, and potentially alterations in the microbiome; thus, further chemical analysis of this data is warranted. 

Taken together, metabolite perturbations may be reflective of increased OS burden and altered antioxidant defenses in premature infants at higher risk of BPD development though additional investigations are needed to further examine the utility of metabolomics in the BPD population.

## 7. Conclusions

BPD, the most common chronic lung disease affecting premature infants, is associated with significant morbidity and mortality. Since BPD was initially described, our clinical, pathophysiological, and histopathological understanding of this disease process has evolved, and we now understand the importance of OS in the pathogenesis of BPD. Here, we have reviewed the factors that increase the susceptibility of preterm infants to OS, including increased OS exposures and an immature antioxidant system. OS plays a pivotal role in the development of BPD by broadly impacting many processes including gene expression, protein function, metabolism—collectively reviewed in the corresponding ‘omics’ sections above. This includes changes in a number of genes related to OS that have been identified by epigenetic and transcriptomic analyses in murine models that mimic BPD and in premature infants at risk of BPD development and infants with established lung disease. Alterations in the genetic regulation of antioxidant enzymes is a common theme in these studies. Metabolomics and proteomics data have also implicated differences in OS-related pathways in BPD patients. Cord blood analysis, though rarely employed in ‘omics’ studies, has the potential to identify early-life biomarkers that are currently lacking in preterm lung disease management. A major limitation to all of these studies is the difficulty of obtaining timely and appropriate samples from human patients. Urinary and fecal metabolites may provide a less invasive methodology to identify patients with a predisposition to develop BPD. Only a few studies on the ‘omics’ of the interaction between OS and BPD have been conducted since 2020 and more studies are needed to further our understanding of the role of OS in the development of BPD, which may prove beneficial for prevention, timely diagnosis, and the development of targeted therapy. 

## Figures and Tables

**Figure 1 antioxidants-11-02380-f001:**
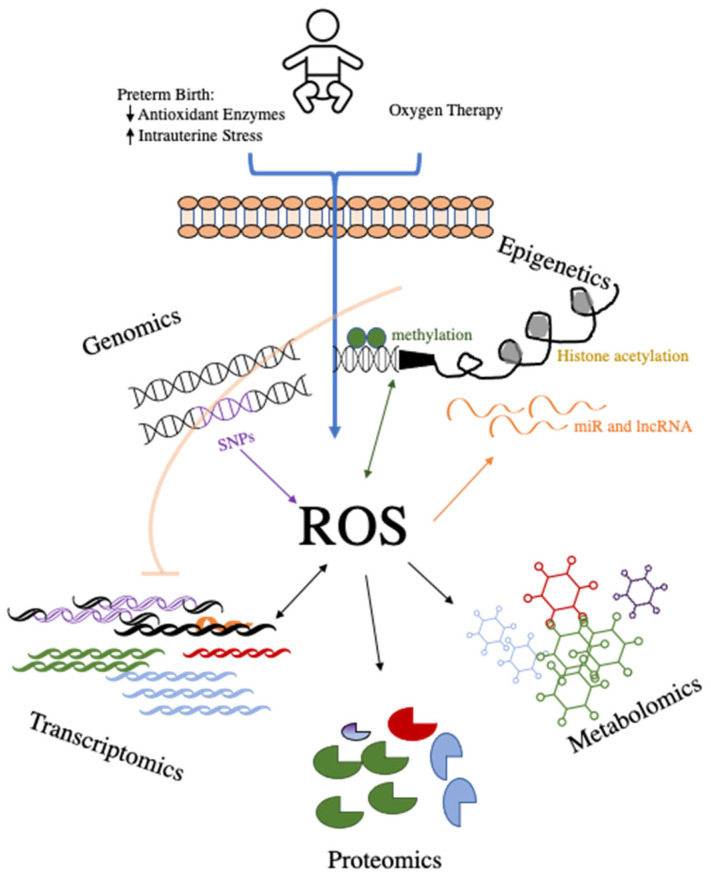
Schematic overview of the interactions of the ‘omics’ and reactive oxygen species (ROS) in preterm infants.

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
