# Peer review of "Pathogenesis of Bronchopulmonary Dysplasia: Role of Oxidative Stress from ‘Omics’ Studies"

_antioxidants, 2022, doi:10.3390/antiox11122380_

Round 1

Reviewer 1 Report

The authors have chosen a very interesting topic for review. However, no convincing effort has been made. It is just summarizing the literature in a careless way.

For example,

In the introduction, there is nothing why this review is important?

What is new in this review?

The authors could explore a bit about reactive nitrogen species (RNS), a very innovative topic.

The title is extensive.

There is only one basic mechanism employed by antioxidant enzymes.

Is there only one conclusion the authors have reached.

There are various grammatical mistakes.

Overall very poorly written.

Reviewer 2 Report

Oxygen therapy has emerged as a critical component of the care of premature infants. The issue is that prolonged exposure to hyperoxia can interfere with the lungs' ability to develop, leading to several diseases, including bronchopulmonary dysplasia (BPD), which is becoming more common yearly. Long-term exposure to high oxygen levels can influence the normal development of lung tissue and vascular beds in premature children leading to BPD. The available research in this area demonstrates that, even though BPD is a disease caused by many different variables, its primary risk factors are early exposure to hyperoxia and the presence of oxidative stress.

In this interesting article, the authors review the results of -omics technologies in BPD with a special focus on oxidative stress, a factor implicated in BPD's pathophysiology. The article is well-written and structured, with an introduction that gradually takes you into the problem. Subsequently, it addresses the results available in each of the -omics, ending with a conclusion appropriate to what has been discussed throughout the review.

In my opinion, the article can be published as long as the authors undertake a series of major modifications that, in my opinion, are important:

1.- When reviewing the bibliography cited in the article, it is striking that it is not up to date; most of the publications are before 2020. A simple search in PubMed with the terms "BPD and oxidative stress and genomics" results in a number of important publications that are not included in this review. The same is true with any of the other -omics. Therefore, the authors should update the bibliography and include the most recent papers in the field.

2.- In addition, the authors should review publication number 90.

Reviewer 3 Report

In this review article, the authors address the question of the extent to which an association between oxidative stress and the pathogenesis of bronchopulmonary dysplasia can be demonstrated. To this end, they consider articles dealing with the relationship between oxidative stress, antioxidants, duration of pregnancy at the time of birth, respiratory problems, and pulmonary pathologies. The authors are examining the effects of altered redox status at multiple levels, starting with the genome, followed by epigenetics, transcription and conversion to proteins, and finally the resulting metabolites.

Because in many cases it is not possible to obtain sufficient experimental data from human studies - for example, by their nature there is no lung material from surviving human children - results from animal studies, e.g., in mice, have been included in this article.

Overall, the article is well written and interesting. However, a few small careless mistakes have crept in, which is why the article should be carefully proofread again.

Round 2

Reviewer 1 Report

Thanks for attending the suggestions. The manuscript has been significantly improved. In view of the above, I believe that the review presents robust and consolidated content.